# Gliadin Nanoparticles Pickering Emulgels for β-Carotene Delivery: Effect of Particle Concentration on the Stability and Bioaccessibility

**DOI:** 10.3390/molecules25184188

**Published:** 2020-09-12

**Authors:** Ce Cheng, Yi Gao, Zhihua Wu, Jinyu Miao, Hongxia Gao, Li Ma, Liqiang Zou, Shengfeng Peng, Chengmei Liu, Wei Liu

**Affiliations:** 1State Key Laboratory of Food Science and Technology, Nanchang University, No. 235 Nanjing East Road, Nanchang 330047, China; 352313318015@email.ncu.edu.cn (C.C.); 412314919015@email.ncu.edu.cn (Y.G.); wuzhihua@ncu.edu.cn (Z.W.); 402313319040@email.ncu.edu.cn (J.M.); 352313319021@email.ncu.edu.cn (H.G.); 402313318047@email.ncu.edu.cn (L.M.); pengsf@ncu.edu.cn (S.P.); liuchengmei@ncu.edu.cn (C.L.); liuwei@ncu.edu.cn (W.L.); 2School of Life Sciences, Nanchang University, Nanchang 330031, China; 3National R&D Center for Freshwater Fish Processing, Jiangxi Normal University, Nanchang 330022, China

**Keywords:** β-carotene, gliadin, Pickering emulgels, rheological property, stability, bioavailability

## Abstract

β-carotene is a promising natural active ingredient for optimum human health. However, the insolubility in water, low oral bioavailability, and instability in oxygen, heat, and light are key factors to limit its application as incorporation into functional foods. Therefore, gliadin nanoparticles (GNPs) Pickering emulgels were chosen as food-grade β-carotene delivery systems. The objectives of the present study were to investigate the influence of GNPs concentration on the rheological properties, stability, and simulated gastrointestinal fate of β-carotene of Pickering emulgels. The formulations of Pickering emulgels at low GNPs concentration had better fluidity, whereas at high GNPs concentration, they had stronger gel structures. Furthermore, the thermal stability of β-carotene loaded in Pickering emulgels after two pasteurization treatments was significantly improved with the increase of GNPs concentration. The Pickering emulgels stabilized with higher GNPs concentration could improve the protection and bioaccessibility of β-carotene after different storage conditions. This study demonstrated the tremendous potential of GNPs Pickering emulgels to carry β-carotene.

## 1. Introduction

β-carotene has the highest provitamin A activity, which is essential for human body, including antioxidant and anti-inflammatory effect, improving the body immunity, and treatment of some diseases such as nyctalopia and xerophthalmia, and is therefore a strong candidate for incorporation into functional foods [1,2]. However, β-carotene is insoluble in water, has low oral bioavailability, and is unstable in oxygen, heat and light due to its multiple conjugate double bonds structure, which can lead to loss of both color and bioactivity of β-carotene in foods and then a decline in product quality and consumer acceptance [3,4]. Generally, the ideal delivery systems should remain stable at sterilization and storage conditions and have the lowest loss of bioactivity until it reached the targeted sites. Besides, due to the limited rate of absorption and transport of the active ingredient at the intestinal epithelial cells, the release rate of active ingredient should be appropriate. Therefore, researchers were committed to develop food-grade delivery systems to improve β-carotene solubility, stability, and bioavailability all the time. A number of effective delivery systems for the encapsulation, protection, and delivery of lipophilic components have been designed, such as emulsions [5], gels [6], liposomes [7], microcapsules [8], etc.

Emulsion gels had the properties of both emulsion and gel based on the high internal phase emulsion or Pickering emulsion template, which typically consisted of either a network of aggregated oil droplets or oil droplets trapped within a biopolymer hydrogel [9,10,11]. Pickering emulsion gels (also named as Pickering emulgels) stabilized with solid colloidal particles as a kind of interface-dominated material were extensively researched as they were prepared free of risky surfactants and had convenient machinability, favorable stability, and good biocompatibility, etc. In Pickering emulgels, colloidal nanoparticles could be strongly adsorbed at oil–water interface, effectively preventing coalescence, Ostwald ripening, drainage, and other unstable pathways, and thus, enhancing the stability of the active components [12]. In recent studies, the role of improving stability of active components through Pickering emulgels prepared by food-grade materials as delivery vehicles, e.g., from proteins, including soy protein [13], zein [14], whey protein [15], pea protein [16], and gliadin protein [17], etc. has been confirmed. The interfacial properties of Pickering emulgels droplet influenced the texture of manufactured food products that was able to specially control the physicochemical stability and the release of functional ingredients, which were fabricated through various approaches including oil fraction, colloidal particles concentration, temperature, pH, ionic strength, extra addition of polysaccharide or protein [9,18], etc. Herein, the change of rheological properties was the most intuitive by adjusting the interfacial properties with colloidal particles or oil concentration. For example, the apparent viscosity of Pickering emulgels stabilized with calcium-cross-linked whey protein nanoparticles was progressively increased with increasing the concentration from 0.2% to 1.0% (*w*/*v*) [15]. The apparent viscosity, storage modulus (G’), and loss modulus (G”) of Pickering emulgels stabilized with ovotransferrin-gum arabic particles were increased as the oil fraction increased from 0.3 to 0.7 [19]. The different rheological properties of food matrices can be satisfied with special designation. Besides the rheological properties, there are also differences in droplet flocculation, formation of a gel network, stability, and gastrointestinal fate at different interfacial properties. For example, Pickering emulgels stabilized with pea protein isolate at pH 3.0 can improve intestinal targeting and sustained release of β-carotene by adjusting oil concentration [20]. However, the influence of colloidal nanoparticles concentration on the gastrointestinal fate of active ingredients has not been reported.

Gliadin is a kind of plant protein with ethanol extracted from wheat, which has an affinity for various hydrophobic compounds and is rich in glutamine and proline residues [21]. As a safe and economically appealing ingredients, and the excellent properties of being metabolizable, biodegradable, and biocompatible make it widely useful as formulating aid in food, drug, and cosmetic industries. Herein, the strong hydrophobicity at the terminals of gliadin led to its low water solubility, which can form nanoparticles through self-assembly process [22]. As early as in 1996, gliadin nanoparticles (GNPs) were prepared as carrier for all-transretinoic acid (RA), and the in vitro release of RA from GNPs exhibited defect of initial burst effect [23]. However, the research showed that the GNPs had the advantage of bioadhesive interactions with the stomach mucosa, which was directly responsible for dramatically increased oral bioavailability of lipophilic drugs [24]. The GNPs had relatively high levels of neutral and lipophilic amino acids, so that they can promote the interaction with mucosa through hydrogen bonding or hydrophobic interactions to obtain excellent bioadhesive behavior, thereby increase the bioavailability of encapsulated lipophilic molecules and prolong their residence time in a certain part of the gastrointestinal [25].

The main objective of present work was to investigate the potential of these Pickering emulgels stabilized with GNPs, acting as delivery systems for lipophilic active ingredients, and better understand the influence of GNPs concentration on the rheological properties, stability, and simulated gastrointestinal fate of β-carotene Pickering emulgels by studying morphologies, rheological properties, thermal stability, storage stability, and gastrointestinal fate. This information will help us to build a food-grade delivery system by Pickering emulsion technology possessing both a stable delivery of lipophilic components and texture controllable Pickering emulgels.

## 2. Results and Discussion

### 2.1. Formation and Microstructure of Pickering Emulgels

In present work, 0.1% β-carotene was dissolved in corn oil (*w*/*w*) and then was directly mixed with GNPs (0.5−1.5%, *w*/*v*) to a final oil fraction of 70% (*w*/*w*). As shown in Figure 1a, the visual observation indicated that all the Pickering emulgels exhibited a bright yellow color. The fresh Pickering emulgels at 0.5% (*w*/*v*) GNPs showed a certain degree of fluidity at an early stage such as semiliquid state (Figure 1b), while the Pickering emulgels changed from liquid to gel-like state and exhibited a state of a plastic gel when the concentration of GNPs was more than 0.75% (*w*/*v*) (Figure 1b). The increase of plasticity might be due to a considerable increase in the amount of colloid particles adsorbed at the oil–water interface [26]. However, it is worth noting that the high concentration of GNPs could enhance the gel strength and gel rate. As shown in the CLSM images of the Pickering emulgels (Figure 3), the oil droplets exhibited a green fluorescence, while the red fluorescence of GNPs was found around the droplets. The droplet size of droplets gradually decreased and arranged more closely with the increasing of GNPs concentration, which was similar with previous result about pea protein Pickering emulsion [16]. This might be due to that the smaller oil droplets provided a greater surface area for adsorption of GNPs and increased interdroplet interactions. It has also been reported that the particle coverage on interface concentration results in a smaller droplet size [27].

### 2.2. Rheological Property of β-Carotene Pickering Emulgels

Understanding the rheological properties of Pickering emulgels are in favor of better processing and utilization of emulgels. Hence, we investigated the effect of GNPs concentration on the rheological properties of GNPs Pickering emulgels. The resulting apparent viscosity versus shear rate data was governed by the Power law model:(1)$$\eta \;=\;k\;{\left(\dot{\gamma }\right)}^{n\;-\;1}$$

Here, η is the apparent viscosity (Pa·s), *k* is the consistency index (Pa·s^n^), $\dot{\gamma \;}$ is the shear rate (s^−1^) and *n* is the flow behavior index. The parameters (η, *k*, and *n*) are presented in Table 1. The apparent viscosity (η) and the consistency index (*k*) increased with increasing concentration of GNPs, which was consistent with the decrease in droplet size and increase in the amount of droplets (as evidenced by the Table 2), leading to the increase in droplet–droplet interactions per unit volume. The increase of GNPs concentration enhances the interaction between proteins adsorbed at the interface of individual oil droplets. As shown in Figure 2a, Pickering emulgels show a typical pseudoplastic fluid flow curve. Apparent viscosity of all the GNPs stabilized Pickering emulgels decreased with the increasing shear rate (Figure 2a), and the *n* values were much less than 1 (Table 1), which exhibited a shear-thinning behavior in the shear rate range of 1–100 s^−1^. This phenomenon was possibly due to the progressive breakdown of the droplet clusters upon the application of increased external shear forces [28]. Moreover, gel-like network of Pickering emulsions can further be confirmed by dynamic frequency oscillatory measurement. As expected, the increase of GNPs concentration enhanced the value of G’ and G’’, and the growth trend of G’ and G’’ gradually slowed down (Figure 2b). In all Pickering emulgels, G’ was one order of magnitude higher than G’’ over the entire frequency range, and both moduli were weakly dependent on frequency. These results indicated that the rheological behavior of Pickering emulgels exhibited typical for a highly flocculated elastic structure.

Temperature-sweep oscillations were measured to evaluate their stability of rheological structure after sterilization. The effect of GNPs concentration on the G′ and the G″ moduli value at high temperature short-term sweep oscillatory measurements is shown in Figure 2c,d. When the temperature is constant at pasteurization condition (90 °C, 3 min and 70 °C, 30 min), the G′ value of the GNPs Pickering emulgels maintained stability and slowly increased during the period of heat preservation, but the G″ value of the GNPs Pickering emulgels showed a fluctuant reduction trend, which led to a significant reduction in the damping factor and indicated the transition from a liquid-like dispersion into a more solid-like gel structure. It can be attributed to further thermal denaturation, such as unfolding and exposing internal hydrophobic groups of protein molecules [29]. Herein, the fluctuation of G’ value might be due to the syneresis of Pickering emulgels at high temperature. However, both moduli value of all samples maintained stability when temperature decreased and renewed to previous tendency.

### 2.3. Thermal and Storage Stability of β-Carotene Pickering Emulgels

Thermal sterilization is necessary for retarding microbial growth in food processing. The microstructure, apparent observations, and β-carotene retention at 70 °C for 30 min and 90 °C for 3 min of Pickering emulsions were determined to understand the differences in thermal stability at various GNPs concentrations. It was shown in Figure 3 that the CLSM images of all Pickering emulgel samples did not vary and were very stable after both pasteurization treatment, which indicated that the GNPs Pickering emulgel has the potential to be a favorable delivery system.

The appearance of Pickering emulgels stabilized by GNPs at various concentrations was still yellow (Figure 4), which indicated that GNPs Pickering emulgels were useful for preventing thermal degradation of β-carotene. The influence of thermal sterilization on volume-weighted mean diameter (d_4,3_) of the Pickering emulgels at various GNPs concentration is shown in Table 2. The results indicated that the value of d_4,3_ progressively decreased with increasing GNPs concentration, and there was no significant difference in d_4,3_ of the Pickering emulgels after thermal sterilization. These results were consistent with the microstructure and visual observations. The impact of pasteurization treatments on retention rate of β-carotene in the Pickering emulgels was also investigated. As shown in Figure 4b, the retention rate of β-carotene in Pickering emulgels was all higher than 92%. The existence of a structured and rigid layer of nanoparticle at the oil–water interface provided an electrostatic barrier against flocculation and coalescence, which could effectively improve the thermal stability of β-carotene. In summary, all these phenomena suggested that the GNPs Pickering emulgels had favorable thermal stability against pasteurization treatments. GNPs Pickering emulgels could be used as a potential carrier system effectively improving the stability of β-carotene, which improves gradually with the increase in GNPs concentration.

Due to its highly unsaturated structure, β-carotene is highly sensitive to high temperature, prooxidants, light, and oxygen during storage [30]. The impact of GNPs concentration on the chemical degradation of β-carotene in Pickering emulgels was investigated after storage at 4, 25, and 55 °C. As shown in Figure 5a, the retention rate of β-carotene stored at 4 °C after 28 days was higher than 80%. The rate of β-carotene degradation increased with temperature, which was in agreement with earlier studies [31]. The relative β-carotene concentration (*C*/*C_0_*) stabilized with 0.5% and 0.75% GNPs decreased from an initial value of 100% to 75.68% and 80.70%, respectively after 28 days storage at 25 °C (Figure 5b) and to 66.67% and 74.41%, respectively after 28 days storage at 55 °C (Figure 5c). However, the retention rate of β-carotene in the Pickering emulgels stabilized with 1% and 1.5% GNPs was still higher than 80% even after 28 days storage at 55 °C. The gliadin has a stronger antioxidant property, which could competitively protect the oil against oxidative damage [32] and further reduce the oxidation of β-carotene. Additionally, the high concentration of GNPs can form firm and inseparable interfacial layers at the oil droplet surface. Thus, the air, light, and diffusion of prooxidants or free radicals might be limited by the thicker coverage of GNPs at interface membrane. The increased density at the oil/water interface might also affect the diffusion of oxygen, free radicals, and prooxidants at interfacial layer and the β-carotene degradation [33,34]. It could be concluded that the storage stability of β-carotene in Pickering emulgels at high GNPs concentrations was greater than that at low GNPs concentrations, which suggested that the GNPs Pickering emulgel was an effective method for stabilizing β-carotene.

### 2.4. Retention and Bioaccessibility of β-Carotene during In Vitro Digestion

Pickering emulgels were potential delivery system for encapsulating and protecting lipophilic bioactive compounds. Thus, the evaluation of gastrointestinal digestive fate had great significance for their further practical application. The stability and bioaccessibility of β-carotene encapsulated in Pickering emulgels at different GNPs concentration were further evaluated using a simulated gastrointestinal tract model. It could be observed in Figure 6a that the gel structure of GNPs Pickering emulgels was destructed by pepsin-hydrolysis after gastric digestion. The Pickering emulgels stabilized with 0.5% GNPs were hydrolyzed thoroughly, and their structure was broken into oil droplets. However, it is worth noting that the hydrolysis rate of Pickering emulgels reduced with the increasing GNPs concentration. This effect can be attributed to the strong gel-like network structure, which could inhibit the destruction of gel state with pepsin and presumably affect the bioaccessibility of β-carotene. The ability of GNPs Pickering emulgels improving the retention and bioaccessibility of β-carotene during intestinal digestion was further evaluated. Part of the digesta was collected to determine the chemical stability of the β-carotene in the Pickering emulgels. The chemical degradation of encapsulated β-carotene in Pickering emulgels decreased with the increasing GNPs concentration (Figure 6b). The degradation of β-carotene in the gastric phase where the pH is 2 could mainly be attributed to their instability at acidic pH. The Pickering emulgels could be designed to improve the β-carotene retention under gastric conditions. The β-carotene easily undergoes protonation, then cis-trans isomerization, and further degradation reactions under acidic conditions. The improved stability of β-carotene in the Pickering emulgels at acidic pH could be due to following reasons: first, the formation of a dense gel network around the oil droplets might increase the pathlength for diffusion of oxygen, prooxidants, and free radicals to the droplet surfaces and second, the antioxidant capacity of the system might increase when high levels of GNPs were incorporated. More antioxidant peptides were released during digestion in simulated GIT [35]. The peptides are known to be natural antioxidants that may inhibit oxidation of bioactive substances. The micelle phase was then collected to determine the bioaccessibility of the β-carotene in the GNPs Pickering emulgels. Interestingly, there was no significant different between samples (shown in Figure 6c). However, the emulgels stabilized with higher concentration of GNPs could be designed to improve β-carotene retention under simulated gastrointestinal conditions as mentioned above. The final amount of β-carotene in the mixed micelle phase of the Pickering emulgels was increased by increasing GNPs concentration.

## 3. Materials and Methods

### 3.1. Materials

Wheat gluten (85%) was purchased from Xunxian Tianlong Flour Co., Ltd. (Hebi, China). Corn oil was purchased from Yihai Kerry Grain and Oil Food Company (Nanchang, China). β-carotene (with purity ≥ 95%) and Coomassie brilliant blue G-250 were obtained from Aladdin Industrial Corporation (Shanghai, China). The enzymes used, including mucin (from porcine stomach), pepsin (from porcine gastric mucosa), lipase (from porcine pancreas), and pancreatin (from porcine pancreas), bile extract, Nile Blue A (N0766), and Nile red (72485) were obtained from the Sigma Chemical Company (St. Louis, MO, USA). All other reagent chemicals used were of analytical grade.

### 3.2. Preparation of Gliadin Nanoparticles

GNPs was prepared according to the antisolvent procedure combining with DHPM treatment method as described in our previous work [10] with few modifications. Precisely, 12 g of gliadin was dissolved in 50 mL anhydrous alcohol using a magnetic stirrer held at room temperature overnight and then centrifuged (5000 rpm, 10 min) to remove the insoluble materials. Subsequently, the gliadin solution was added into 150 mL bulk water and stirred continuously to form coarse particles of gliadin molecules. The resulting GNPs dispersion was prepared by passing the coarse particles twice through a high-pressure microfluidizer (M-110EH30, Microfluidic Corp., Newton, MA, USA) at 12,000 psi. Acetic acid was removed by dialysis against deionized water in a dialysis bag (8–14 kDa molecular weight cut-off, Solarbio, Beijing, China) until the pH value of the GNPs was 5.5. The gliadin concentration in the nanoparticles solutions was determined using Coomassie brilliant blue (G-250) method as described previously [36].

### 3.3. Preparation of β-Carotene Pickering Emulgels

β-carotene (0.01%, *w*/*w*) was dissolved in the corn oil using a magnetic stirrer held at 90 °C for 5 min, and the resulting mixtures were dispersed with an acoustic ultrasonic generator (SG3200HET, Shanghai Gutel Ultrasonic Instrument Co., Ltd., Shanghai, China) for 2 min. The above steps were repeated for three times. After that, the sample was immediately cooled in an ice bath to room temperature before use. For the preparation of β-carotene Pickering emulgels, the β-carotene mixtures were directly mixed with different concentrations of GNPs (diluted to 0.5−1.5%, *w*/*v*) to a final oil phase fraction of 70% (*w*/*w*). The mixtures (with a total weigh of 20 g) were prehomogenized using a high-speed dispersing homogenizer (ULTRA-TURRAX^®^ T18 digital, IKA, Staufen, Germany) with 19 mm dispersion head operating at 12,000 rpm for 2 min.

### 3.4. Droplet Size Analysis

Droplet size of β-carotene Pickering emulgels was measured by Malvern laser particle size analyzer (MS3000, Malvern Instruments Ltd., Worcestershire, UK). The fresh samples were diluted with phosphate buffer (10 mM, pH 5.5) prior to analysis to avoid multiple scattering effects. The particle size was reported as the volume-weighted mean diameter (d_4,3_).

### 3.5. Microstructure Observation and Rheology Measurement

#### 3.5.1. Microstructure Observation

The microstructures of the Pickering emulgels were observed by confocal laser scanning microscopy (CLSM) according to a previous description [37]. Various specimens of the Pickering emulgels were dyed by adding Nile red to the oil phase and Nile blue A to the water phase before emulsion preparation. The dyed specimens were placed into a glass bottom cell culture dish (φ 20 mm) and observed under CLSM (Carl Zeiss LSM710, Jena, Germany). The fluorescent dyes were excited by either an argon laser at 488 nm for Nile Red or a helium neon (He-Ne) laser at 633 nm for Nile Blue A.

#### 3.5.2. Rheology Measurement

The static and dynamic rheological measurements of emulgels were determined using a Shear Rheometer (MCR302, Anton Paar, Germany) at 25 °C. About 2 mL of fresh Pickering emulgels was poured onto the plate and waited for 10 min to allow thermal equilibrium before rheological experiment. The gap between two plates was set to 1.0 mm. The steady shear measurements were increased from 0.1 to 100 s^−1^, and the data were reported as apparent viscosity. The dynamic frequency sweeping was oscillated from 1 to 100 rad/s, and the variations of storage modulus (G’) and loss modulus (G”) were recorded. The above test data were collected using a plate-and-plate geometry (pp-50, 50 mm diameter). Moreover, the oscillatory temperature sweep measurement was measured using a concentric cylinder geometry Couette (CC27, 26.6 mm diameter and 40.0 mm measuring height) held at 70 °C for 30 min and 90 °C for 3 min, respectively. The heating and cooling temperature rates were maintained at 5 °C/min at a fixed frequency of 1 Hz and strain of 1%, and the variations of G’ and G” were also recorded.

### 3.6. Measurement of Thermal and Storage Stability of β-Carotene Pickering Emulgels

The thermal stability of β-carotene Pickering emulgels was determined by pasteurizations under two different conditions according to a previous description [38]. Pasteurizations were performed in a water bath at 70 °C for 30 min and 90 °C for 3 min, respectively. The storage stability of β-carotene Pickering emulgels was tested by storing them at 4, 25, and 55 °C separately in the dark for 28 days. The thermal and storage stabilities of β-carotene contents were measured as described in Section 3.7.4.

### 3.7. In Vitro Digestion

The gastrointestinal fate of β-carotene Pickering emulgels was simulated by using an in vitro gastrointestinal model consisting of mouth, gastric, and intestinal phases. The parameters selected for the simulation of gastrointestinal conditions were based on previously reported recommendations [39].

#### 3.7.1. Mouth Phase

Simulated saliva fluid (SSF) containing 3% mucin was based on those reported in previous studies [40,41]. A 7.5 g proper dilution of the Pickering emulgels was mixed with 7.5 mL of SSF. The pH of the mixture was adjusted to 6.8 and the emulgels were incubated at 37 °C for 10 min with continuous agitation at 100 rpm.

#### 3.7.2. Gastric Phase

Simulated gastric fluid (SGF) was prepared by adding 2 g NaCl, 7 mL HCl, and 3.2 g of pepsin (from porcine gastric mucosa) in 1 L of distilled water, and the pH was adjusted to 1.2 using 1.0 M HCl. The samples taken from the mouth phase were mixed with SGF (ratio 1:1 *w*/*w*). The pHs of the samples were adjusted to pH 2 using NaOH (1 M) and incubated at 37 °C for 2 h with continuous agitation at 100 rpm.

#### 3.7.3. Small Intestine Phase

The mixtures obtained from the gastric phase were further incubated for 2 h at 37 °C in a simulated small intestinal fluid (SIF) containing 2.5 mL pancreatic lipase (4.8 mg mL^−1^), 4 mL bile extract solution (5 mg mL^−1^), and 1 mL calcium chloride solution (750 mM). The volume of NaOH (0.1 M) was added to maintain a constant pH of 7.0 within the reaction chamber during 2 h of small intestine digestion period.

#### 3.7.4. Determination of β-Carotene Bioaccessibility and Stability

At the end of the GIT model, the raw digesta was centrifuged at 10,000 g for 30 min (Anting TGL-16B, Shanghai, China, 1.5 ML×12), and then, the middle micelle phase was collected. The extracted β-carotene was collected and analyzed by an ultraviolet–visible spectrophotometer at 450 nm. The concentration of β-carotene was determined using a preprepared calibration curve. The retention index and bioaccessibility of β-carotene were measured according to the equation described in previous studies [42].
Retention index = 100 × C_Digesta_/C_Initial_(2)
Bioaccessibility = 100 × C_Micelle_/C_Digesta_(3)

Here, C_Initial_ is the concentration of β-carotene in the emulsion, C_Micelle_ is the concentration of β-carotene in the mixed micelle phase, and C_Digesta_ is the concentration of β-carotene in the total digesta collected after the small intestine phase.

### 3.8. Statistical Analysis

All data were analyzed using SPSS Version 18.0 software, and a least significant difference (LSD) with a confidence interval of 95% was applied to compare the means. Data are presented as mean ± standard deviation (SD).

## 4. Conclusions

In conclusion, the rheological properties, stability, and simulated gastrointestinal fate of Pickering emulgels encapsulated β-carotene at different GNPs concentration were investigated. The plasticity and rheological behavior of these Pickering emulgels could be modulated according to the concentration of GNPs. All of the GNPs Pickering emulgels after two pasteurization treatments exhibited high β-carotene retention, and the higher GNPs concentration further slowed down the degradation of β-carotene and change in rheological properties of GNPs Pickering emulgels. Meanwhile, it was noteworthy that the increasing concentration of GNPs results in improved β-carotene stability in Pickering emulgels at various temperature storage conditions. The result of the simulated gastrointestinal digestion indicated that the higher GNPs concentration provided a strong gel structure, which could promote the transport of β-carotene into the micellar layer. The GNPs Pickering emulgels exhibited a great potential to be a sustained release delivery system of β-carotene. This research has provided some useful insights for the rational design of Pickering emulgels-based delivery systems to encapsulate, protect, and deliver liposoluble components in foods and other commercial products.

## Figures and Tables

**Figure 1 molecules-25-04188-f001:**
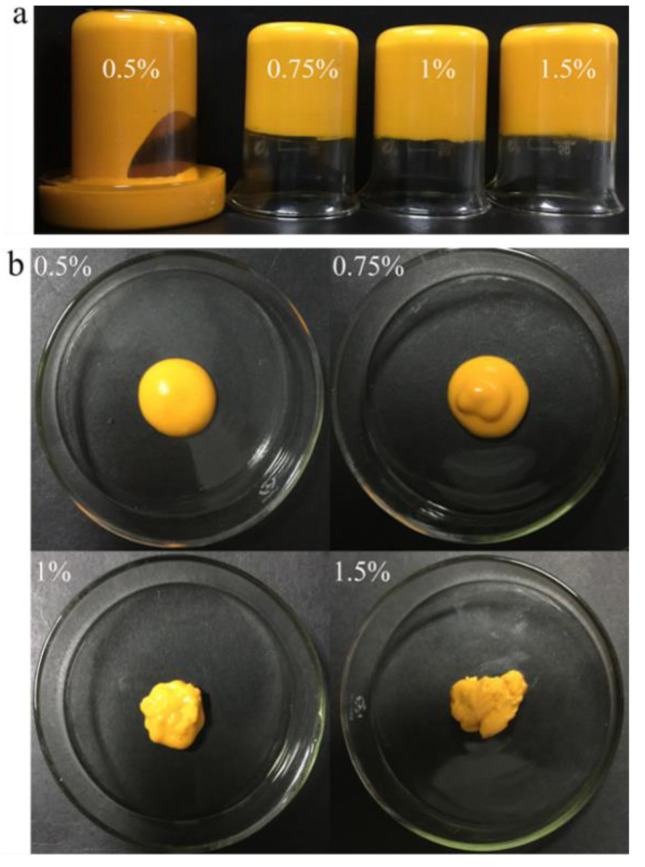
The visual images of Pickering emulgels at different concentrations of gliadin nanoparticles (GNPs) (0.5%−1.5%, *w*/*v*) in (**a**) beaker and (**b**) culture dish.

**Figure 2 molecules-25-04188-f002:**
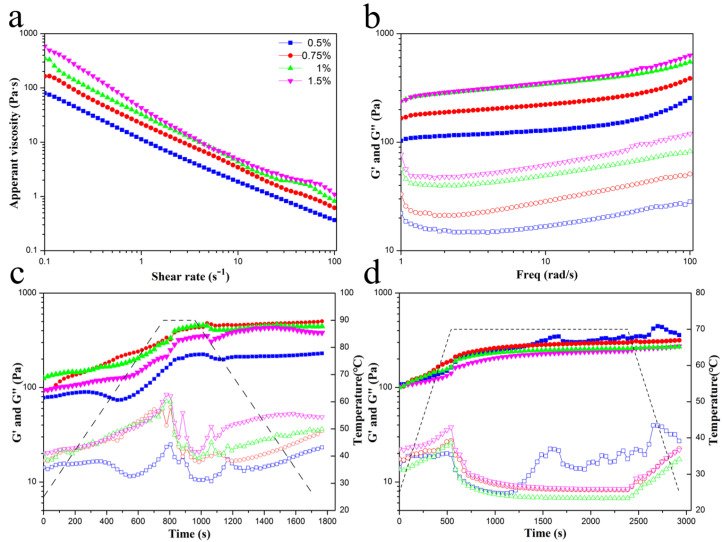
Rheological properties of GNPs Pickering emulgels at different concentrations (0.5%−1.5%, *w*/*v*). (**a**) Apparent viscosity versus shear rate, (**b**) oscillatory frequency sweep curves, (**c**) oscillatory higher temperature short-term sweep measurement curves held at 90 °C for 3 min; (**d**) oscillatory lower temperature long-term sweep measurement curves held at 70 °C for 30 min. Solid and hollow symbols denote G′ and G″, respectively.

**Figure 3 molecules-25-04188-f003:**
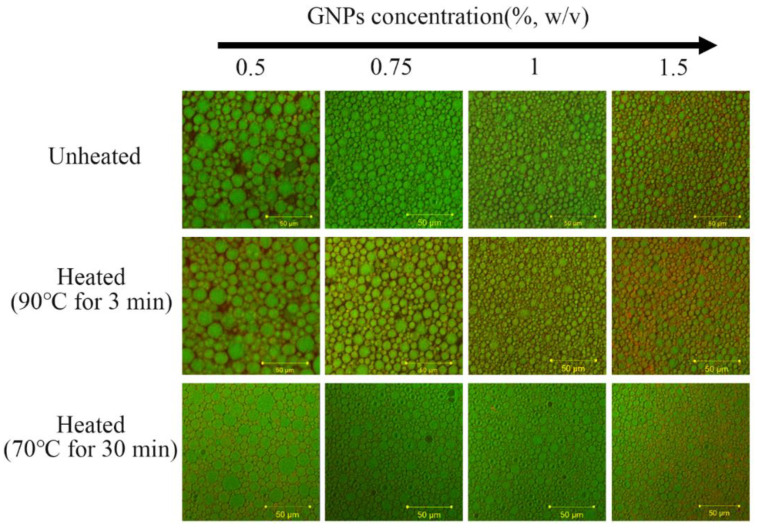
The CLSM images of GNPs Pickering emulgels at room temperature, 90 °C for 3 min and 70 °C for 30 min.

**Figure 4 molecules-25-04188-f004:**
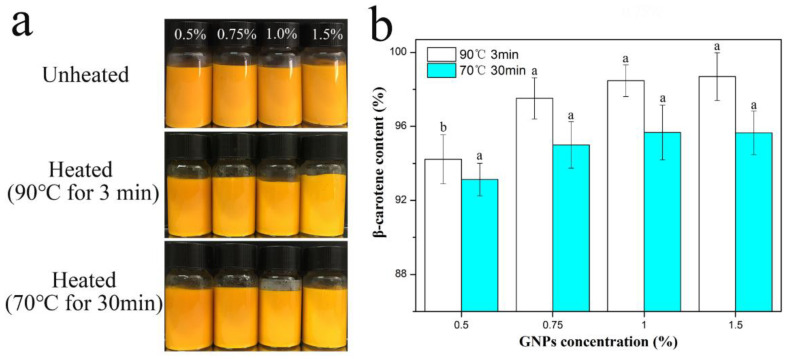
(**a**) The visual images and (**b**) β-carotene retention of GNPs Pickering emulgels at room temperature and pasteurizations conditions.

**Figure 5 molecules-25-04188-f005:**
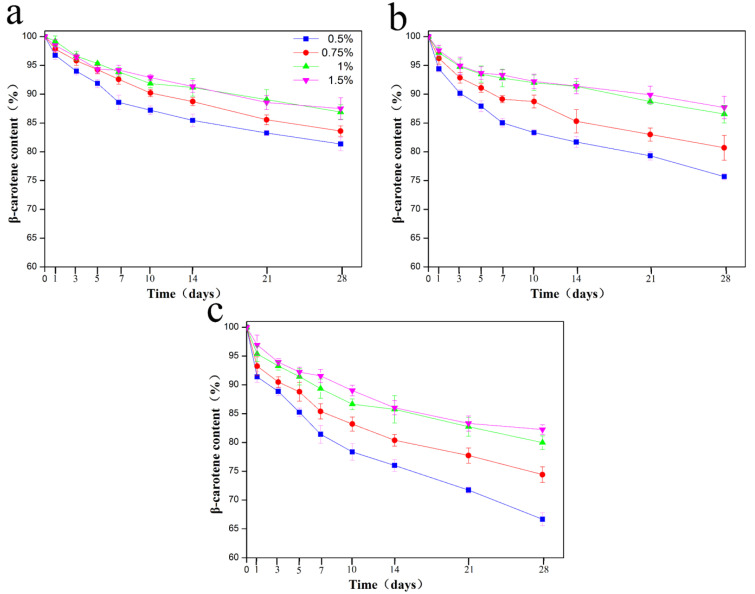
Effect of the concentration of GNPs Pickering emulgels on the β-carotene storage stability at (**a**) 4 °C, (**b**) 25 °C, and (**c**) 55 °C for 28 days.

**Figure 6 molecules-25-04188-f006:**
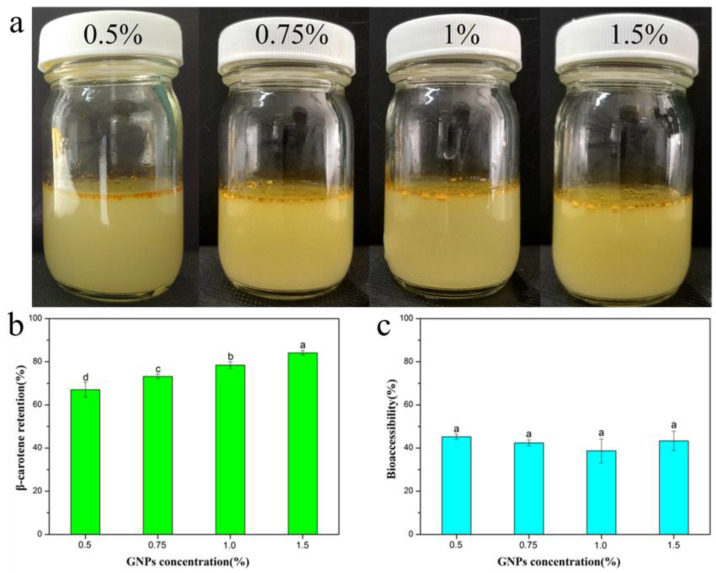
(**a**) Visual images of Pickering emulgels after the simulated gastric digestion, (**b**) β-carotene retention, and (**c**) bioaccessibility of GNPs Pickering emulgels during the simulated intestinal digestion.

**Table 1 molecules-25-04188-t001:** The apparent viscosity (η), consistency index (*k*), and flow behavior index (*n*) of Pickering emulgels with gliadin nanoparticles (GNPs) concentration.

GNPs Concentration (%)	η	*k*	*n*	R^2^
0.5	0.365 ± 0.012 ^d^	11.989 ± 0.085 ^d^	0.214 ± 0.001 ^a^	0.998
0.75	0.614 ± 0.015 ^c^	23.478 ± 0.492 ^c^	0.181 ± 0.012 ^b^	0.998
1	0.818 ± 0.024 ^b^	34.860 ± 1.245 ^b^	0.156 ± 0.026 ^b^	0.993
1.5	1.090 ± 0.041 ^a^	50.018 ± 0.746 ^a^	0.088 ± 0.008 ^c^	0.982

Apparent viscosity is the value at shear rate equals to 100 s^−1^. Samples designated with different letters (a, b, and c) were significantly different (Duncan, *p* < 0.05).

**Table 2 molecules-25-04188-t002:** Mean diameter of β-carotene varied with GNPs concentration (mean ± SD, *n* = 3).

GNPs Concentration (%)	d_4,3_ (μm)		
	Unheated	90 °C 3 min	70 °C 30 min
0.5	11.34 ± 0.46 ^b^	13.78 ± 0.45 ^a^	13.69 ± 0.43 ^a^
0.75	9.71 ± 0.52 ^c^	11.37 ± 0.33 ^b^	10.99 ± 0.46 ^b^
1	7.53 ± 0.48 ^d^	7.67 ± 0.36 ^d^	7.81 ± 0.28 ^d^
1.5	4.94 ± 0.28 ^e^	4.91 ± 0.29 ^e^	5.09 ± 0.36 ^e^

Samples designated with different letters (a, b, and c) were significantly different (Duncan, *p* < 0.05).

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
