# Peer review of "Gliadin Nanoparticles Pickering Emulgels for β-Carotene Delivery: Effect of Particle Concentration on the Stability and Bioaccessibility"

_molecules, 2020, doi:10.3390/molecules25184188_

Round 1

Reviewer 1 Report

The manuscript entitled “Gliadin nanoparticles Pickering emulgels for β-carotene delivery: Effect of particle concentration on the stability and bioaccessibility” is an appreciable piece of work. Indeed, beside describing the preparation and the physico-chemical characterization of the Pickering emulsion gels loaded with β-carotene, the authors take into consideration the stability of β-carotene in the preparations, not only to the thermal treatments for pasteurization and to storage but also in simulated gastrointestinal fluids, aiming at devising systems for the efficient delivery of lipophilic molecules.

The weak point of the text are the many grammar and syntax mistakes, which hamper the reading. In my opinion extended English editing is required prior to publication.

Minor details:

the pictures of of Figure 1c are the same in the first row of Figure 3. They might be deleted from Figure 1 c, making reference to Figure 3 (Unheated samples) in the text.

ref. 10 at the publisher site the years is 2018 instead of 2017

Author Response

The manuscript entitled “Gliadin nanoparticles Pickering emulgels for β-carotene delivery: Effect of particle concentration on the stability and bioaccessibility” is an appreciable piece of work. Indeed, beside describing the preparation and the physico-chemical characterization of the Pickering emulsion gels loaded with β-carotene, the authors take into consideration the stability of β-carotene in the preparations, not only to the thermal treatments for pasteurization and to storage but also in simulated gastrointestinal fluids, aiming at devising systems for the efficient delivery of lipophilic molecules.

- We thank the reviewer for their positive comments on our manuscript.

The weak point of the text are the many grammar and syntax mistakes, which hamper the reading. In my opinion extended English editing is required prior to publication.

- We have already done the English corrections before submitting the manuscript.

Minor details:

the pictures of of Figure 1c are the same in the first row of Figure 3. They might be deleted from Figure 1 c, making reference to Figure 3 (Unheated samples) in the text.

- As suggested, we have deleted this picture from Figure 1c.

ref. 10 at the publisher site the years is 2018 instead of 2017

- As suggested, we have revised the year on the publisher’s site.

Reviewer 2 Report

The work is interesting, there are many works about pickering emulgel, but it is interesting and innovative for the combination with beta carotene.

For my skills I found the work interesting.

There are some spelling errors:

Lane 70: on underlined

107: results_about

176: against underlined

The spss 18.0 statistics program is now quite obsolete, but for the data processed it can still be fine.

Author Response

The work is interesting, there are many works about pickering emulgel, but it is interesting and innovative for the combination with beta carotene.

For my skills I found the work interesting.

- We thank the reviewer for their positive comments. As suggested, we have revised the manuscript according to following suggestions.
There are some spelling errors:

Lane 70: on underlined

- As suggested, this spelling errors has been revised (line 69).

107: results_about

- As suggested, this spelling errors has been revised (line 110).

176: against underlined

- As suggested, this spelling errors has been revised (line 177).

The spss 18.0 statistics program is now quite obsolete, but for the data processed it can still be fine.

- We thank the reviewer for their valuable comments, we will use more advanced SPSS statistics program to process data in our future research.

Reviewer 3 Report

The overall quality of this manuscript is acceptable. Important conclusions can be supported by experimental results. I suggest publishing this manuscript after a revision, although some specific comments are list below.

1- A revision of references is required. Some references are old and there are many recent and relevant studies with emulgels, zein pickering emulsions and beta-carotene encapsulation that should be cited.

2- Flow curves may be fitted to modified power-law model (see for example "Trujillo-Cayado, L. A., Natera, A., García González, M. D. C., Muñoz, J., & Alfaro Rodríguez, M. D. C. (2015). Rheological properties and physical stability of ecological emulsions stabilized by a surfactant derived from cocoa oil and high pressure homogenization.").

Author Response

The overall quality of this manuscript is acceptable. Important conclusions can be supported by experimental results. I suggest publishing this manuscript after a revision, although some specific comments are list below.

- We thank the reviewer for their positive comments. As suggested, we have revised the manuscript according to following suggestions.

1- A revision of references is required. Some references are old and there are many recent and relevant studies with emulgels, zein pickering emulsions and beta-carotene encapsulation that should be cited.

- As suggested, we have revised the reference according to your comments.

2- Flow curves may be fitted to modified power-law model (see for example "Trujillo-Cayado, L. A., Natera, A., García González, M. D. C., Muñoz, J., & Alfaro Rodríguez, M. D. C. (2015). Rheological properties and physical stability of ecological emulsions stabilized by a surfactant derived from cocoa oil and high pressure homogenization.").

- As suggested, we have added relevant content according to your comments (line 118-131).

Round 2

Reviewer 3 Report

I consider that the manuscript can be published in present form.